# Effect of Oxygen Nonstoichiometry on Electrical Conductivity and Thermopower of Gd_0.2_Sr_0.8_FeO_3−δ_ Ferrite Samples

**DOI:** 10.3390/ma12010074

**Published:** 2018-12-26

**Authors:** Vyacheslav Dudnikov, Yury Orlov, Aleksandr Fedorov, Leonid Solovyov, Sergey Vereshchagin, Alexander Burkov, Sergey Novikov, Sergey Ovchinnikov

**Affiliations:** 1Kirensky Institute of Physics, Federal Research Center Krasnoyarsk Scientific Center, Russian Academy of Sciences, Siberian Branch, 660036 Krasnoyarsk, Russia; slad63@yandex.ru (V.D.); jso.krasn@mail.ru (Y.O.); sgo@iph.krasn.ru (S.O.); 2Institute of Engineering Physics and Radio Electronics, Siberian Federal University, 660041 Krasnoyarsk, Russia; 3Institute of Chemistry and Chemical Technology, Federal Research Center Krasnoyarsk Scientific Center, Russian Academy of Sciences, Siberian Branch, 660036 Krasnoyarsk, Russia; leosol@icct.ru (L.S.); snv@icct.ru (S.V.); 4Ioffe Institute, 194021 St. Petersburg, Russia; a.burkov@mail.ioffe.ru (A.B.); snpochta@gmail.com (S.N.)

**Keywords:** rare-earth-substituted cobalt oxides, stability, thermoelectric properties

## Abstract

The behavior of the resistivity and thermopower of the Gd_0.2_Sr_0.8_FeO_3−δ_ ferrite samples with a perovskite structure and the sample stability in an inert gas atmosphere in the temperature range of 300–800 K have been examined. It has been established that, in the investigated temperature range, the thermoelectric properties in the heating‒cooling mode are stabilized at *δ* ≥ 0.21. It is shown that the temperature dependencies of the resistivity obtained at different *δ* values obey the activation law up to the temperatures corresponding to the intense oxygen removal from a sample. The semiconductor‒semiconductor electronic transitions accompanied by a decrease in the activation energy have been observed with increasing temperature. It is demonstrated that the maximum thermoelectric power factor of 0.1 µW/(cm·K^2^) corresponds to a temperature of T = 800 K.

## 1. Introduction

The major characteristics of a thermoelectric material are its thermoelectric efficiency *Z* = *S*^2^*σ*/*ϰ* (*S* is the Seebeck coefficient, also known as thermopower, *σ* is the electrical conductivity, and ϰ is the thermal conductivity) [1], dimensionless thermoelectric quality factor *ZT* (*T* is the absolute temperature), and power factor *P* = *S*^2^*σ* = *S*^2^/*ρ* (*ρ* is the resistivity). 

Recently, close attention of researchers has been paid to oxide materials [2], since they are stable against high temperatures, nontoxic, and, as a rule, do not contain rare elements. A group of complex oxides of the rare-earth transition metals Ln_1−x_A_x_MO_3−δ_ (Ln is a lanthanoid, A is an alkali or alkali-earth metal, and M is a transition metal (Co, Fe, or Mn)) with a perovskite structure holding a special place [3,4,5] among the oxides for promising application. The diverse physicochemical properties of these compounds, which belong to strongly correlated electron systems, are interesting both for application [6] and for fundamental research. Along with their thermoelectric characteristics, the stability of these materials against different environmental factors, including the atmospheric composition and temperature, is of great practical importance. In this study, taking into account the significant differences between the low-temperature properties of the Gd_1−x_Sr_x_FeO_3−δ_ compounds with different oxygen nonstoichiometry indices *δ* [7] and the high mobility of oxygen in the Gd_1−x_Sr_x_СoO_3−δ_ compounds [8], we investigate the effect of the oxygen nonstoichiometry on the thermoelectric properties of the Gd_0.2_Sr_0.8_FeO_3−δ_ compound and its stability in the temperature range of 300–800 K.

## 2. Materials and Methods 

Sample synthesis. The Gd_0.2_Sr_0.8_FeO_3−δ_ samples were obtained using the glycinenitrate-based solution-combustion synthesis [9]. The initial components of the perovskiteoxide were cations of nitrates of each metal component (Gd(NO_3_)_3_·6H_2_O, 99.9% trace metal basis, Rare Metals Plant, Novosibirsk, Russia; Sr(NO_3_)_2_, >99.0%, Reachim, Moscow, Russia; Fe(NO_3_)_3_·9H_2_O, >98%, Sigma-Aldrich, St. Louis, MO, USA). The initial gel was obtained via evaporation of the glycinenitrate-based solution at 363 K with subsequent heating until combustion at a low rate. The prepared powder was calcinated in air at 723 K for 5 h, thoroughly grinded in anagate mortar, and sintered in air at 1623 K for 5 h with subsequent cooling to room temperature at a rate of 2 °C/min. Tablets were synthesized at *T*_c_ = 1473 K for 24 h and cooled in the same mode. 

The oxygen nonstoichiometry of the samples was determined using a NETZSCHSTA449C analyzer equipped with an Aёolos QMS 403C mass spectrometer (NETZSCH, Selb, Germany). The measurements were performed in the 5% H_2_-Ar mixture flow in a corundum crucible with a perforated cover. The sample mass was 30 mg. The oxygen content in the Gd_0.2_Sr_0.8_FeO_3−δ_ samples was determined from the mass decrement ∆*m* using the equation *δ*(*T*) = *δ*_0_ + *M∆m*(*T*)/1600, where *δ*(*T*) is the *δ* value at temperature *T*, *∆m*(*T*) (%) is the mass decrement at this temperature, and *M* is the molecular mass of the Gd_0.2_Sr_0.8_FeO_2.87_ compound. The measurement error was δ = ±0.01. The oxygen nonstoichiometry index of the initial Sr_0.8_Gd_0.2_FeO_3−δ_ material was taken to be *δ*_0_ = 0.13, according to the data on the Gd-Sr-Fe perovskite synthesized using the same technique [7].

The X-ray diffraction measurements were performed on powder diffractometer (PANalytical X’Pert PRO, Eindhoven, The Netherlands) using Cu *K*α radiation in the 2θ-angular range of 20–158°. The crystal structure was refined using the derivative difference minimization (DDM) method [10]. Temperature dependences of the thermopower and resistivity were obtained on an experimental setup developed at the Ioffe Institute, St. Petersburg [11,12].

## 3. Results

In contrast to the complex cobalt oxides Re_1−x_Sr_x_CoO_3−δ_ (Re is the rare-earth ion), which, depending on the cooling rate and synthesis temperature, can form compounds ordered or disordered over А sites of the perovskite structure [13,14,15,16,17,18], the Gd_0.2_Sr_0.8_FeO_3−δ_ compound synthesized by us is disordered and characterized by the random distribution of oxygen vacancies and cations over the А sites. The X-ray diffraction study revealed no foreign phases. According to the X-ray powder diffraction data (Figure 1), the synthesized Gd_0.2_Sr_0.8_FeO_3−δ_ samples, similar to the heavily-doped La_1−x_Sr_x_FeO_3−δ_ (0.8 ≤ x ≤ 1.0) compounds [19], have a form of a disordered perovskite with the cubic symmetry (sp. gr.Pm3m (a = 3.8688(1) Å), in which all the iron sites were identical, as in the mixed-valence iron compounds Sr_2_LaFe_3_O_8.94_ [20]. The structural formula derived from the refined occupancies of atomic positions (Table 1) was Sr_0.8_Gd_0.2_FeO_2.79_. Similar perovskites Sr_0.75_Gd_0.25_FeO_2.87_ and Sr_0.75_Gd_0.25_FeO_2.94_ described in [7] have smaller lattice parameters (3.8665(1) and 3.8582(1) Å), which is consistent with the smaller oxygen stoichiometry estimated from the XRD refinement for our sample.

Atomic positions, and their occupancies and thermal parameters in the crystal structure of Sr_0.8_Gd_0.2_FeO_2.79_ are given in Table 1.

Figure 2 shows the temperature-programmed reduction data. The change in the mass caused by removal of oxygen from the sample upon heating in the 5% H_2_-Ar reducing atmosphere started at ≈500 K and depended on the heating rate, which is related to the solid-state reduction kinetics. As the heating rate increased, the thermogravimetric (TG) and derivative thermogravimetric (DTG) curves shifted to the high-temperature region and the maximum reduction rate was observed at 640 and 725 K at heating rates of 2 and 20 K/min, respectively.

The electrical resistivity and thermopower were measured on the rectangular bar samples 0.9 × 4.5 × 9 mm in size at temperatures from 300 to 800 K in the inert He (99.999%) atmosphere. To examine the changes in the temperature dependences of the resistivity and thermopower, the measurements were performed during the heating–cooling cycles and the courses of the obtained temperature curves were compared. Three thermal cycles were performed. The heating and cooling rates were 5 °C/min. The results obtained are shown in Figure 3. It can be seen that the temperature dependences of the electrical resistivity and thermopower of the investigated samples in the heating‒cooling cycles were essentially different, which is indicative of the strong effect of oxygen on the physical properties of the Gd_0.2_Sr_0.8_FeO_3-δ_ compound. In the first thermocycle, when oxygen was actively removed from the sample, the behavior of the thermoelectric parameters of the compound was especially anomalous (Figure 4а). For clarity, the electrical resistivity and Seebeck coefficient measured during the first heating with the active oxygen removal (Figure 4а) and the last cooling (Figure 4b), when the sample was almost stable, are plotted. 

The anomalies in the behavior of the Seebeck coefficient correlated with the data of thermogravimetric analysis of the sample mass loss caused by the removal of oxygen and are observed at much lower temperatures than in the behavior of the resistivity (Figure 4a). 

The oxygen nonstoichiometry index δ increased from one heating‒cooling cycle to another and the deviation from the semiconductor-type conductivity shifted toward higher temperatures. At δ = 0.21, the temperature dependency of the resistivity was qualitatively consistent with the semiconductor-type conductivity dρ/dT < 0 over the entire temperature range of interest (Figure 4b). The conductivity in the regions corresponding to the semiconductor type obeyed the thermal activation law:(1)$$\rho \left(T\right)=\rho_{0}\times exp\left(E_{a}/k_{B}T\right)$$
where $\rho_{0}$ is the coefficient weakly dependent on temperature, $E_{a}$ is the activation energy, and $k_{B}$ is the Boltzmann constant. The dependences of the logarithmic resistivity of the sample on the reciprocal temperature in the corresponding temperature ranges for several heating and cooling processes are presented in Figure 5. For all the thermal cycles, the ln(ρ)(1/T) curves contain two portions that are described well by the thermal activation law with different activation energies $E_{a}$. At each specific δ value, the activation energy decreased with increasing temperature. At the same time, the activation energy grew with the oxygen nonstoichiometric index. The absolute values of the oxygen nonstoichiometry indices and activation energies for different temperature ranges and electronic transition temperatures were determined by the crossing points of the approximation curves are given in Table 2. 

Figure 6 presents the plots of activation energy $E_{a}$ in the low-temperature (below the temperature of the electronic transition, T < T_p-p_) and high-temperature (above the transition temperature, T > T_p-p_) regions and semiconductor‒semiconductor transition temperature T_p-p_ as functions of the oxygen content in the sample. It can be seen that, in the investigated temperature and δ ranges, all the dependencies are linear. 

The $E_{a}$ value in the low-temperature range was several times higher than in the high-temperature range; the difference slightly decreased with increasing δ. The electrical conductivity of the materials under study was implemented using impurity carriers induced by different-valence Gd+3/Sr+2 cation substitutions and holes related to oxygen vacancies. At high temperatures, we observed the properties of an almost degenerate semiconductor, while at low temperatures, the activation energy increased. The temperature T_p-p_ of the asymptotic change weakly depended on the vacancy concentration. The fact that the activation energy depended on the vacancy concentration is evidence for the shift of the impurity level from the allowed state band extremum. It is worth noting that the concentration dependences of the activation energies (straights 1 and 2 in Figure 6) are almost parallel.

Figure 7 shows the temperature dependencies of the power factor *P* for the first and last temperature cycles. It can be seen that, as the temperature increased, the power factor of the samples grew, but did not attain its maximum, in the investigated temperature range. In the sample stability region, the increase was monotonic, without jumps, and almost linear for the sample with δ = 0.21. At a temperature of T = 500 K, the power factor of the sample with δ = 0.21 exceeded almost fourfold the power factor of the samples with δ = 0.13 (0.052 µW/(K^2^·cm) and 0.013 µW/(K^2^·cm)) and the maximum *P* value obtained at T = 800 K was 0.1 µW/(K^2^·cm).

The power factors obtained in this work for the sample Gd_0.2_Sr_0.8_FeO_3-δ_ (*P* = 0.1 μW/(cm·K^2^)) are consistent with those presented in Reference [21] for the LaCo_1−x_Ni_x_O_3_ (*P* = 0,12 μW/(cm·K^2^), LaCo_1−x_Ti_x_O_3−δ_ (*P* = 0.28 μW/(cm·K^2^)) [22], La_1−x_Sr_x_Co_0.8_Ni_0.1_Fe_0.1_O_3_ (*P* = 0.76 μW/(cm·K^2^)) [5], and La_1−x_Na_x_CoO_3_ (*P* = 0.1 μW/(cm·K^2^)) polycrystalline samples [23], although our data are inferior to the value of *P* ≈ 3 μW/(cm·K^2^) for the Ca_3−x_Bi_x_Co_4_O_9+δ_ samples from study [24]. 

## 4. Conclusions

The behavior of electrical resistivity and thermopower of the disordered Gd_0.2_Sr_0.8_FeO_3−δ_ perovskite in the temperature range of 300–800 K and its stability in the helium atmosphere were investigated. It was established that the stability of thermoelectric parameters strongly depends on the oxygen content in a sample and is obtained at temperatures from 300 to 800 K at an oxygen nonstoichiometric index of δ ≥ 0.21.

In the sample stability region, the temperature dependences of electrical resistivity for all δ values contained portions that were described well by the thermal activation law; as the temperature increased, spread semiconductor‒semiconductor transitions were observed, which are accompanied by a decrease in the activation energy. The latter grows with decreasing oxygen content in a sample. In this case, the temperature of electronic transition depends linearly on the oxygen nonstoichiometry. 

We explained the nature of the semiconductor transitions as the result of occupancy/deoccupancy by the oxygen ions of different positions in the nonstoichiometry lattice. The electron bands near the Fermi level substantially changed and led to significant changes in the band gap and the activation energy. The activation energies in the low- and high-temperature regions depended similarly on the δ value. The increase in the δ value led to the significant enhancement of the power factor *P*. In particular, at a temperature of *T* = 500 K, the power factor of the sample with δ = 0.21 was almost fourfold higher than the power factor of the samples with δ = 0.13 (0.052 µW/(K^2^·cm) and 0.013 µW/(K^2^·cm). In the sample with δ = 0.21, the power factor increased with temperature almost linearly and attained its maximum value of 0.1 µW/(cm·K^2^) at *T* = 800 K. Thus, varying the oxygen content in a sample, one can control the temperature ranges of stability and the power factor of the complex transition metal oxides.

## Figures and Tables

**Figure 1 materials-12-00074-f001:**
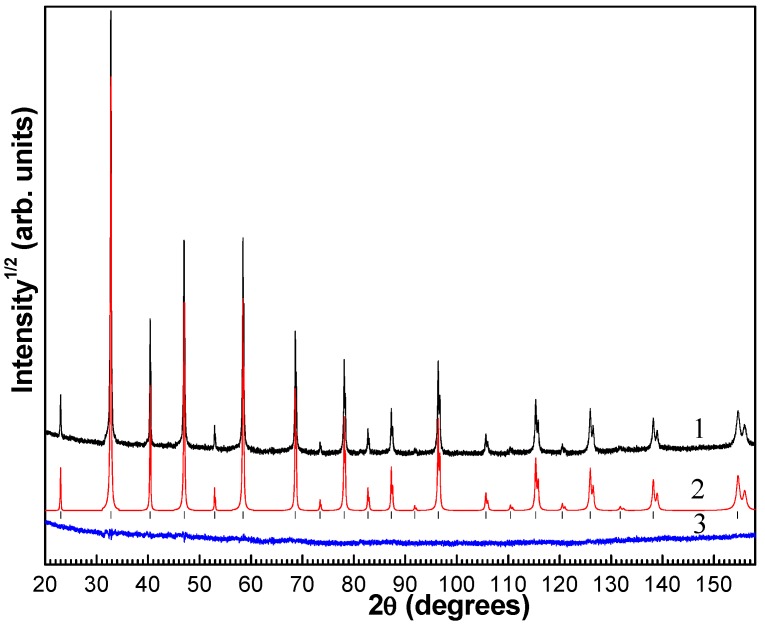
Experimental (1), calculated (2), and difference (3) X-ray diffraction patterns of the Gd_0.2_Sr_0.8_FeO_2.79_ compound after DDM refinement.

**Figure 2 materials-12-00074-f002:**
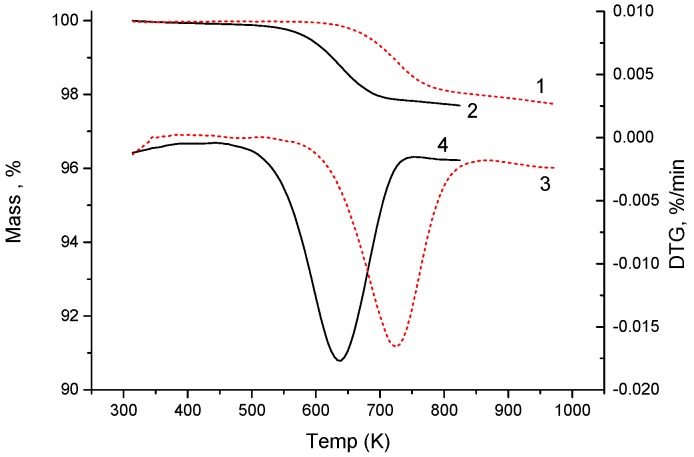
(1, 2) TG and (3, 4) DTG curves of temperature-programmed reduction of the Gd_0.2_Sr_0.8_FeO_2.87_ compound in the 5% H_2_-Ar mixture flow at heating rates of 2 (black solid curve) and 20 K/min (red dotted curve).

**Figure 3 materials-12-00074-f003:**
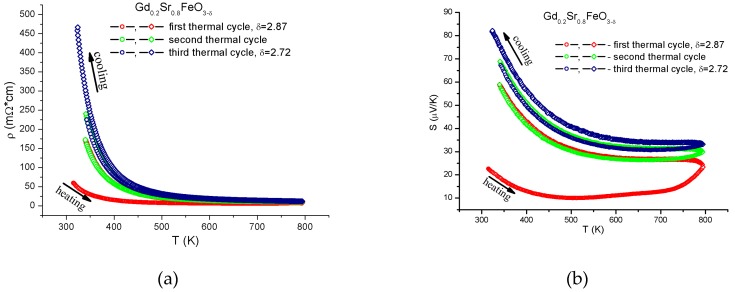
Temperature dependences of (**а**) the resistivity and (**b**) thermopower for three continuous heating‒cooling cycles.

**Figure 4 materials-12-00074-f004:**
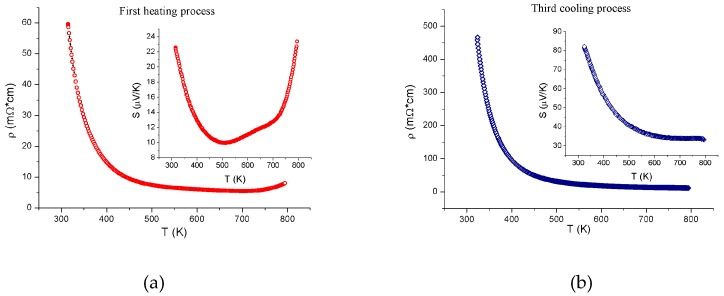
Temperature dependences of the resistivity and thermopower (insets) during (**a**) the first heating and (**b**) the third cooling.

**Figure 5 materials-12-00074-f005:**
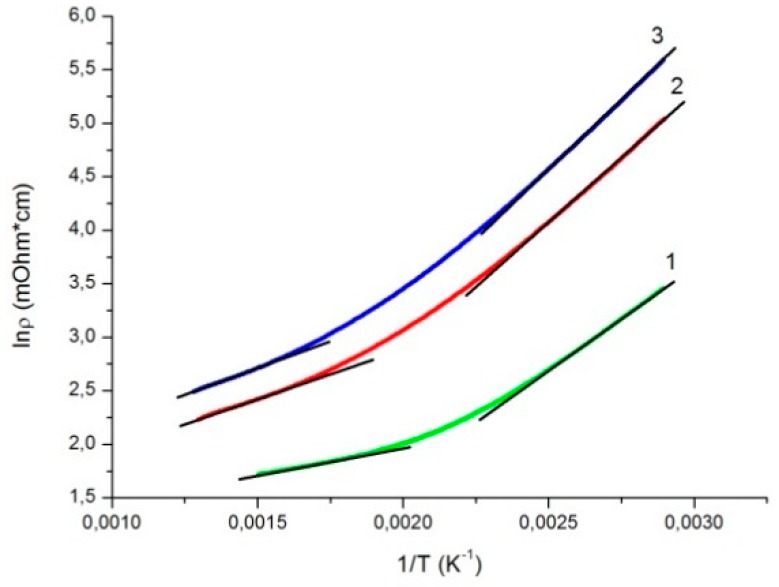
Reciprocal temperature dependence of the Gd_0.2_Sr_0.8_FeO_3−δ_ logarithmic resistivity. Black straight lines show the thermal activation law processing: (1) first heating, (2) first cooling, and (3) third cooling.

**Figure 6 materials-12-00074-f006:**
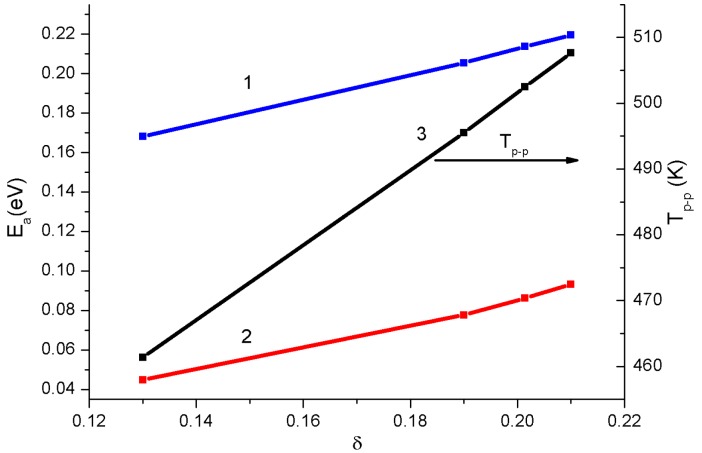
Dependencies of the activation energy $E_{a}$: (1) is the temperature region below the electronic transition and (2) is the high-temperature region, and (3) semiconductor–semiconductor transition temperature T_p-p_ on the oxygen nonstoichiometry index.

**Figure 7 materials-12-00074-f007:**
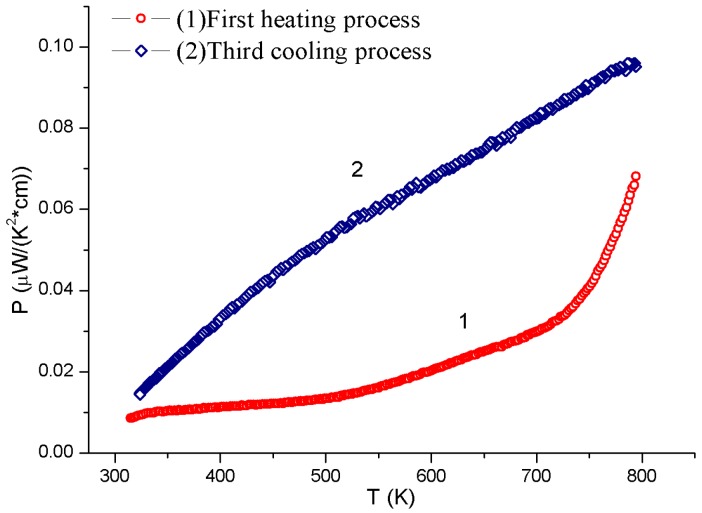
Temperature dependences of the power factor *P* for the first heating (δ = 0.13 at T = 300 K) (1) and the third cooling (δ = 0.21) (2).

**Table 1 materials-12-00074-t001:** Atomic positions, and their occupancies and thermal parameters in the crystal structure of Sr_0.8_Gd_0.2_FeO_2.79_. Space group is Pm3m, with lattice parameter a = 3.8688(1) Å.

Atom	Position	Occupancy	x	y	z	U_iso_ [Å^2^]
Sr	1a	0.800(4)	0	0	0	0.0092(7)
Gd	6e	0.0334(7)	0.080(4)	0	0	0.0092(7)
Fe	1b	1	1/2	1/2	1/2	0.0082(5)
O	12h	0.232(1)	0.452(2)	1/2	0	0.0211(14)

**Table 2 materials-12-00074-t002:** Oxygen nonstoichiometry indices, activation energies, and electronic transition temperatures for the Gd _0.2_Sr_0.8_FeO_3−δ_ ferrite sample.

Regime	δ	E_a_ (T < T_p-p_) (eV)	T_p-p_ (K)	E_a_ (T > T_p-p_) (eV)
**First heating**	0.13 ± 0.01	0.16 ± 0.01	461 ± 1	0.04 ± 0.01
**First cooling**	0.19 ± 0.01	0.20 ± 0.01	495 ± 1	0.07 ± 0.01
**Second cooling**	0.20 ± 0.01	0.21 ± 0.01	502 ± 1	0.08 ± 0.01
**Third cooling**	0.21 ± 0.01	0.22 ± 0.01	507 ± 1	0.09 ± 0.01

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
