# Peer review of "Effect of Oxygen Nonstoichiometry on Electrical Conductivity and Thermopower of Gd0.2Sr0.8FeO3−δ Ferrite Samples"

_materials, 2018, doi:10.3390/ma12010074_

Round 1

Reviewer 1 Report

The paper is well organized and clear. I would suggest to add comparison with literature data, also of other similar materials, in order to compare thermoelectric power of this compound with respect to state of art materials.

Then, some attemp to explain the nature of the semiconductor-semiconductor transition, indicated by the change of Ea with temperature, should be done. What is the main change in the defects structure which is causing such a transition?

Author Response

Dear reviewer!

Thank you very much  for your advices and useful criticism. In compliance with your remarks and suggestions we made changes to the corresponding places of the text and figures.

Comment 1. The paper is well organized and clear. I would suggest to add comparison with literature data, also of other similar materials, in order to compare thermoelectric power of this compound with respect to state of art materials.

Answer 1. Relevant information and the references have been added to the article.

The power factors obtained in this work for the sample Gd 0.2Sr0.8FeO3-δ  (Р = 0.1 μW/(cm·K2)) are consistent with those presented in study [1] for the LaCo1-xNixO3 (Р = 0,12 μW/(cm·K2), LaCo1-хTiхO3-δ (P = 0.28 μW/(cm·K2)) [2], La1-xSrxCo0.8Ni0.1Fe0.1O3 (P = 0.76 μW/(cm·K2)) [3], and La1-xNaxCoO3 (P = 0.1μW/(cm·K2)) polycrystalline samples [4], although our data are inferior to the value of P ≈ 3 μW/(cm · K2) for the Ca3-xBixCo4O9+δ samples from study [5].

[1] R. Robert, M.H. Aguirre, L. Bocher, M. Trottmann, S. Heiroth, T. Lippert, M. Döbeli, A.

Weidenkaff. Thermoelectric Properties of LaCo1-xNixO3 Polycrystalline Samples and Epitaxial Thin

Films. Solid State Sciences, 10 (2008), pp. 502–507.

[2] R. Robert, L. Bocher, M. Trottmann, A. Reller, A. Weidenkaff. Synthesis and High-Temperature Thermoelectric Properties of Ni and Ti Substituted LaCoO3. Journal of Solid State Chemistry, 179 (2006), pp. 3893–3899.

[3] S.G. Harizanova, E.N. Zhecheva, V.D. Valchev, M.G. Khristova, R.K. Stoyanova. Improving the Thermoelectric Efficiency of Co Based Ceramics. Materials Today: Proceedings, 2 (2015), pp. 4256–4261.

[4] S. Behera, V.B. Kamble, S. Vitta, A.M. Umarji, C. Shivakumara. Synthesis Structure and Thermoelectric Properties of La1-xNaxCoO3 Perovskite Oxides. Bulletin of Materials Science, 40(7) (2017), pp. 1291–1299.

D. Moser, L. Karvonen, S. Populoh, M. Trottmann, A. Weidenkaff. Influence of the Oxygen Content on Thermoelectric Properties of Ca3-xBixCo4O9+δ System. Solid State Sciences, 13 (2011), pp. 2160–2164.

Comment 2. Then, some attempt to explain the nature of the semiconductor-semiconductor transition, indicated by the change of Ea with temperature, should be done.

Answer 2.

We explain the nature of semiconductor-semiconductor transitions  as the result of occupancy/ deoccupancy by the oxygen ions of different positions in the nonstoichiometry  lattice. At that the electron bands near the Fermi level  substantially changed and  leads to a significant changes in the band gap and the activation energy respectively.

Comment 3. What is the main change in the defects structure which is causing such a transition?

Answer 3.

See the previous explanation

Reviewer 2 Report

Dear Editor

I have reviewed the  manuscript entitled

„Effect of Oxygen Nonstoichiometry on the Electrical Conductivity and Thermopower

of the Gd 0.2Sr0.8FeO3-δ Ferrite Samples”

Thank you for sending this article for reviewing by me.

In my opinion the manuscript need major revision.

Best regards

Line 28 Mathematical formulas should be numbered throughout the article.

Line 60 The authors should confirm the composition of  Gd 0.2Sr0.8FeO3-δ by measuring content Gd:Sr.

 Line 156 Power factor is an important parameter conditioning the application of  Gd 0.2Sr0.8FeO3-δ.  In my opinion Authors should compare the power factor of  Gd 0.2Sr0.8FeO3-δ with similar materials in the literature and demonstrate the advantage of the synthesized material over others.

Author Response

Dear reviewer!

Thank you very much  for your advices and useful criticism. In compliance with your remarks and suggestions we made changes to the corresponding places of the text and figures.

Comment 1. Line 28 Mathematical formulas should be numbered throughout the article.

Answer 1. It is corrected.

Comment 2. Line 60 The authors should confirm the composition of  Gd 0.2Sr0.8FeO3-δ by measuring content Gd:Sr.

Answer 2.

Gd0.2Sr0.8FeO3-d powders were prepared by the glycine–nitrate process starting from Gd(NO3)3·6H2O (99.9% trace metal basis, Rare Metals Plant, Russia), Sr(NO3)2 (>99.0% trace metal basis, Reachim, Russia), Fe(NO3)3·9H2O (>98% trace metal basis, Sigma-Aldrich).

The sol-gel synthesis applied excludes conditions which may be a cause of the selective cation removal from the charge mixture (during gelation, ignition or calcinations).  Since the purity of Gd/Sr nitrates were at least >99% (trace metal basis) and the initial substances (about 5 g) were weighed with 0.1 mg accuracy,  the Gd/Sr-ratio in the resulting mono-phase sample may be considered to be equal to one in the charge mixture.

The text was corrected respectively.

Comment 3. Line 156 Power factor is an important parameter conditioning the application of  Gd 0.2Sr0.8FeO3-δ.  In my opinion Authors should compare the power factor of  Gd 0.2Sr0.8FeO3-δ with similar materials in the literature and demonstrate the advantage of the synthesized material over others.

Answer 3. Relevant information and the references have been added to the article.(Reviewer 1, answer 1)

Reviewer 3 Report

Interesting article. useful research results.

i reccomend to publish, however the following things can be clarified Before publication.

structural study by XRD can be complemented with neutron diffraction. Even with PDF analysis to reveal more on local structures. there seems signficant evidence of annomalies in the fit.

Neutron can reveal more on O occupancies as well.

2. Is the materials structure  already in the structural databases? please comment on and compare on similar materials. More information of structural analysis is significant for the material.

3. Why not above 800K measured and is the material stable above 800 K?

4. Why Helium mediam is used not clarified.

5. Why P in fig. 7 for 0.13 is steeper above 700k than 0.21? Are there any previous results on delta of above 0.5? is it possible to comment on the limit of delta values withn this series of materials?

Author Response

Dear reviewer!

Thank you very much  for your advices and useful criticism. In compliance with your remarks and suggestions we made changes to the corresponding places of the text and figures.

Comment 1. Structural study by XRD can be complemented with neutron diffraction. Even with PDF analysis to reveal more on local structures. There seems signficant evidence of annomalies in the fit. Neutron can reveal more on O occupancies as well.

Answer 1. We do not have an opportunity for a neutron diffraction study, but we performed more comprehensive XRD structure analysis that clarifies the reviewer's inquiries. The O-position occupancy was refined, from which the oxygen nonstoichiometry was estimated and compared with that in similar
materials.

Comment 2. Is the materials structure  already in the structural databases? Please comment on and compare on similar materials. More information of structural analysis is significant for the material.

Answer 2. A table with structural parameters has been added to the article.

Comment 3. Why not above 800K measured and is the material stable above 800 K?

Answer 3.

Measurements in the higher temperatures range lead to rapid wear of the measuring contacts, the replacement of which is associated with some difficulties. Therefore, in our measurements we restrict ourselves to this temperature range.

An increase in oxygen nonstoichiometry leads to an increase in the temperature range of stability.

Comment 4. Why Helium media is used not clarified.

Answer 4. Measurement in a helium atmosphere does not lead to oxidation of the measuring contacts and shows more stable measurement results. This contributes to longer thermocouples operation.

Comment 5. Why P in fig. 7 for 0.13 is steeper above 700k than 0.21? Are there any previous results on delta of above 0.5? Is it possible to comment on the limit of delta values with this series of materials?

Answer 5. Figure 7 has been corrected (Fig. 7. shows the change in the power factor during the first heating (δ = 0.13 at T = 300 K) and the last cooling (δ = 0.21). As a result of the first heating, there is an active release of oxygen from the sample, which explains the course of the temperature dependence. In the case of the last cooling, the sample is almost stable.

We did not examine our composition for maximum oxygen non-stoichiometry and we did not encounter studies with a δ > 0.5.

Round 2

Reviewer 2 Report

I have reviewed the  manuscript entitled

„Effect of Oxygen Nonstoichiometry on the Electrical Conductivity and Thermopower

of the Gd 0.2Sr0.8FeO3-δ Ferrite Samples”

In my opinion the manuscript can be accepted in present form.